# Design and Realization of a Novel Robotic Manta Ray for Sea Cucumber Recognition, Location, and Approach

**DOI:** 10.3390/biomimetics8040345

**Published:** 2023-08-04

**Authors:** Yang Liu, Zhenna Liu, Heming Yang, Zheng Liu, Jincun Liu

**Affiliations:** 1National Innovation Center for Digital Fishery, China Agricultural University, Beijing 100083, China; liuyang951852682@163.com (Y.L.); hemingng@gmail.com (H.Y.); liuzheng1414@163.com (Z.L.); 2Key Laboratory of Smart Farming Technologies for Aquatic Animals and Livestock, Ministry of Agriculture and Rural Affairs, China Agricultural University, Beijing 100083, China; 3Beijing Engineering and Technology Research Centre for Internet of Things in Agriculture, Beijing 100083, China; 4College of Information and Electrical Engineering, China Agricultural University, Beijing 100083, China; 5Shandong Labor Vocational and Technical College, Jinan 250022, China; liu_zhenna@163.com

**Keywords:** underwater bionic manta ray, mechanism design, motion control, underwater object detection, underwater robot visual perception, underwater robot target approaching

## Abstract

Sea cucumber manual monitoring and fishing present various issues, including high expense and high risk. Meanwhile, compared to underwater bionic robots, employing autonomous underwater robots for sea cucumber monitoring and capture also has drawbacks, including low propulsion efficiency and significant noise. Therefore, this paper is concerned with the design of a robotic manta ray for sea cucumber recognition, localization, and approach. First, the developed robotic manta ray prototype and the system framework applied to real-time target search are elaborated. Second, by improved YOLOv5 object detection and binocular stereo-matching algorithms, precise recognition and localization of sea cucumbers are achieved. Thirdly, the motion controller is proposed for autonomous 3D monitoring tasks such as depth control, direction control, and target approach motion. Finally, the capabilities of the robot are validated through a series of measurements. Experimental results demonstrate that the improved YOLOv5 object detection algorithm achieves detection accuracies (mAP@0.5) of 88.4% and 94.5% on the URPC public dataset and self-collected dataset, respectively, effectively recognizing and localizing sea cucumbers. Control experiments were conducted, validating the effectiveness of the robotic manta ray’s motion toward sea cucumbers. These results highlight the robot’s capabilities in visual perception, target localization, and approach and lay the foundation to explore a novel solution for intelligent monitoring and harvesting in the aquaculture industry.

## 1. Introduction

The high economic value of sea cucumber products has led to the rapid development of sea cucumber aquaculture [1,2]. During the sea cucumber farming process, real-time recognition and localization of sea cucumbers play a vital role in monitoring their growth status and facilitating the capture of farmed sea cucumbers. Currently, underwater manual operations are the primary means for sea cucumber monitoring and harvesting. However, prolonged underwater operations pose significant risks to personnel due to factors such as high pressure and low-temperature [3]. Therefore, highly intelligent autonomous underwater robots offer convenience for underwater mobile monitoring and harvesting [4,5]. The traditional autonomous underwater robots are commonly driven by propellers during underwater operations. They are prone to entanglement with aquatic vegetation and suffer from disadvantages such as low propulsion efficiency and high noise, which cause significant disturbance to aquatic organisms [6]. In contrast, fish species have evolved physiological structures and functional characteristics adapted to their survival environment through long-term natural evolution. Therefore, bio-inspired underwater robots, which mimic biological morphological structures and locomotion mechanisms, exhibit advantages such as high maneuverability, low noise, high stability, and efficiency [7]. To enhance sea cucumber farming efficiency, as well as monitoring and harvesting efficiency, and reduce the environmental interference caused by underwater operations, it is necessary to design a highly maneuverable bio-inspired underwater robot with visual perception capabilities. By integrating high-precision object detection algorithms and edge computing, real-time monitoring, recognition, and localization of sea cucumber health can be achieved, thus further reducing labor costs and operational risks in the sea cucumber aquaculture industry.

Although fish species vary in terms of their types and body shapes, their swimming patterns can be primarily categorized into two modes based on the source of propulsion: Body and/or Caudal Fin (BCF) mode and Median and/or Paired Fin (MPF) mode [8]. The BCF mode primarily generates thrust through undulations of the body and oscillations of the caudal fin, while the MPF mode utilizes the undulations of the pectoral fins, pelvic fins, and other fin surfaces to provide propulsion. Therefore, the BCF mode excels in speed compared to the MPF mode. The MPF mode combines high propulsion efficiency, maneuverability, and stability [9], enabling agile maneuvers such as low-speed turning and rapid acceleration. Robotic fish using the BCF mode inevitably exhibit lateral body movements during swimming, which significantly affect the quality of image capture. On the other hand, the MPF mode demonstrates superior disturbance resistance, making it more suitable for underwater mobile monitoring platforms equipped with cameras and other electro-optical sensors. In nature, the manta ray’s swimming motion is often compared to bird flight, representing a typical example of the MPF mode [10]. These motivate researchers and engineers to create bio-inspired designs that can outperform current state-of-the-art underwater robots in maneuverability and stability. The robotic manta ray, characterized by its agile swimming, gliding capabilities, and exceptional stability, is an ideal biomimetic model for underwater robots equipped with various electro-optical sensors to perform agile underwater tasks. It provides a stable mobile platform for diverse underwater activities.

Extensive research on underwater bio-inspired robotic fish based on the manta ray had been conducted both domestically and internationally. As early as 2002, Davis [11] from Columbia University designed a biomimetic pectoral fin prototype using Shape Memory Alloy (SMA) as a linear actuator. With further research, numerous manta ray prototypes have been developed based on different design principles, considering their motion performance and external morphology [7]. These prototypes include pectoral fins with simple structures, fewer degrees of freedom (DOFs), and high skeletal rigidity. For instance, the RoMan-I prototype from Nanyang Technological University utilized motor-driven rigid fin rays as pectoral fins [12]. Beihang University developed the Robo-ray I-III, which incorporated carbon fiber as the material for the pectoral fins [13,14,15,16]. Some researchers had also focused on flexible pectoral fins that mimic morphological characteristics. However, these designs lack propulsive force. For example, the Institut Supérieur de Mécanique de Paris had developed a miniature flexible robotic manta ray using Dielectric Elastomer Minimum Energy Structures (DEMES) material [17]. Considering the impact of manta ray size on application scenarios, larger robotic manta rays exhibit superior performance and reliability, making them suitable for carrying large-scale sensing equipment. However, these large-scale designs were characterized by complex structures, high costs, larger dimensions, and reduced stealth capabilities, making them less suitable as underwater monitoring platforms for aquaculture. In contrast, smaller robotic manta rays offer advantages such as smaller size, simple structures, lower costs, and higher stealth capabilities, which enhance their engineering practicality. For example, Northwestern Polytechnical University developed a maneuverable robotic manta ray using a zigzag spring support structure, enabling it to perform maneuvers with arbitrary radii [18]. In terms of visual perception, researchers have also conducted corresponding studies on manta rays. The Automation Institute of the Chinese Academy of Sciences developed a robotic manta ray equipped with a visual system and proposed an algorithmic framework for real-time digital video stabilization [19]. Northwestern Polytechnical University achieved manta ray relative positioning by combining improved target detection algorithms and binocular distance measurement using a robotic manta ray equipped with dual cameras [20]. The fusion of visual perception and deep learning techniques in robotic fish will be a future development trend for underwater bio-inspired robots.

With the advancement of deep learning and edge computing technologies, combined with lightweight image processing algorithms, robotic fish with visual perception capabilities can achieve real-time online processing of image data. Object detection is an important means of visual perception for underwater robotic fish. Convolutional neural network-based object detection algorithms can be divided into two-stage and one-stage algorithms. Two-stage algorithms, mainly represented by the RCNN series [21,22,23], achieve higher detection accuracy but have slower processing speeds. One-stage algorithms, mainly represented by the YOLO series [24,25,26,27] and SSD series [28,29], have faster inference speeds. In recent years, with the advantages of Transformer in global feature extraction, it has been successfully applied to dense prediction tasks [30,31]. For example, the Swin Transformer [32] constructed a pyramid structure with gradually decreasing resolutions to realize feature learning based on the Transformer at multiple scales and extract short-range and long-range visual information. Experimental results demonstrated the superiority of this algorithm. Exploring lightweight and high-precision object detection algorithms to be embedded in bio-inspired robotic manta rays is particularly important for enhancing their visual perception capabilities. Additionally, localization algorithms based on binocular vision and semi-global block matching (SGBM) [33,34] will provide stereo visual perception capabilities for underwater robots.

Therefore, the objective of this paper is to design and implement a small bio-inspired manta ray with visual perception capabilities and a rigid-flexible coupled pectoral fin. It aims to enable sea cucumber recognition, localization, and approach, thus establishing the foundation for monitoring the activity status of sea cucumbers and subsequent automated harvesting. The main contributions can be summarized as follows:Designing a novel robotic manta ray with visual perception capabilities and a rigid-flexible coupled pectoral fin.Improving the YOLOv5s object detection and incorporating binocular stereo-matching algorithms to achieve accurate sea cucumber identification and localization.Designing a fuzzy PID controller to realize depth control, direction control, and target approach motion control for the robotic manta ray.

The remaining structure of this paper is as follows: Section 2 elaborates on the overall electromechanical design of the rigid-flexible coupled pectoral fin bio-inspired manta ray. In Section 3, the sea cucumber recognition and localization algorithm based on the improved YOLOv5s object detection and SGBM binocular stereo matching is introduced. Section 4 focuses on the depth control, direction control, and approach motion control of the manta ray based on localization information. Experimental results of the sea cucumber recognition and localization algorithm, as well as the depth control, direction control, and approach motion control of the manta ray, are presented in Section 5. Section 6 provides a discussion of the research presented in this paper. Finally, Section 7 concludes the entire paper with a comprehensive summary.

## 2. Overview of Robotic Manta Ray

The manta ray, as a typical fish utilizing the MPF mode of propulsion, exhibits outstanding stability and maneuverability during motion [35]. It also demonstrates remarkable agility and disturbance resistance at low speeds, making it highly suitable for carrying various optoelectronic sensors and performing flexible maneuvers underwater. The undulatory fins of the manta ray inspire the propulsor design of the robotic manta ray.

To ensure the integrity and consistency of the bio-inspired robotic manta ray, a top-down design approach is employed for the mechanical structure design. First, the overall shape of the bio-inspired robotic manta ray is designed from a holistic perspective. Second, considering the practical requirements, functionalities, performance, and constraints of the entire system, the bio-inspired robotic manta ray is decomposed into three separate sub-components: pectoral fins, caudal fin, and body shell. Finally, employing a local design approach, each component module with different functionalities is gradually refined and designed.

The bio-inspired robotic manta ray operates underwater in a marine environment; therefore, the materials used must possess characteristics such as lightweight, high strength, corrosion resistance, good plasticity, and ease of processing [36]. Considering the compressive strength and corrosion resistance of the resin [37], the black resin is chosen for constructing the body shell of the robotic manta ray. This paper analyzes the shape characteristics of manta rays based on the propulsion mode of manta rays in nature and knowledge from biomimetics. The mechanical structure of the bio-inspired robotic manta ray is rationally simplified. Based on this analysis, the design parameters for the caudal fin, rigid body shell, and rigid-flexible coupled pectoral fins are determined. Figure 1a illustrates the overall rendering of the bio-inspired robotic manta ray, and Figure 1b shows the prototype of the bio-inspired robotic manta ray. Table 1 provides the technical parameters of the bio-inspired robotic manta ray.

### 2.1. Internal Layout of Robotic Manta Ray

The rigid shell of the robotic manta ray provides ample space for accommodating various electronic devices, control components, and batteries. The internal layout, as shown in Figure 2, includes four sets of 7.4 V lithium batteries positioned at the central bottom of the shell to lower the center of gravity and ensure balance. Above the battery compartment, the controller, inertial measurement unit (IMU), and battery level monitoring module are placed at a relatively higher position to protect the electronic components from direct damage in case of accidental water ingress. The attitude sensor is centrally located within the internal space of the shell, accurately capturing the manta ray’s posture. The power module is connected to a separate battery compartment through support pillars at the bottom of the shell, providing both convenience of connection and waterproofing functionality. The machine vision computing module, equipped with a Jetson Xavier NX board, is located at the back of the robotic manta ray, powered by a dedicated 14.8 V battery. The two buoyancy balance units, positioned on both sides of the robotic manta ray, serve to adjust the center of gravity, thereby increasing stability and balancing buoyancy forces.

The bottom layout of the robotic manta ray is depicted in Figure 3. The waterproof electric switch, charging port, and depth sensor are positioned within the central groove of the robotic manta ray. This design can avoid affecting the overall hydrodynamic performance. The binocular camera, as shown in Figure 3b, is externally mounted on the bottom of the robotic manta ray, facilitating easy disassembly and expansion.

### 2.2. Pectoral Fin Undulation Design

The pectoral fin is the most crucial locomotion organ of the manta ray [38] and serves as the core design element in the robotic manta ray. According to relevant biological research, the complex and flexible deformation of the pectoral fin during stable cruising can be decomposed into the superposition of two orthogonal traveling waves [39]. As shown in Figure 4, traveling wave I propagate from the base to the tip of the pectoral fin along the span direction, while traveling wave II approximately propagates from the head to the tail along a chord parallel to the water flow. By coordinating these two sets of traveling waves, the manta ray achieves efficient and agile motion.

Inspired by this, this paper proposes a bio-inspired manta ray pectoral fin design scheme, where the propulsion mechanism of the pectoral fin employs a simple configuration of two pairs of fin strips and a flexible membrane wing. The overall structure of the pectoral fin is illustrated in Figure 5a. Each pectoral fin is equipped with two digital servos capable of continuous bidirectional rotation from 0 to 180 degrees, enabling independent or synchronized control. This design scheme allows for switching between undulating and flapping propulsion modes. The servo motion of the pectoral fin follows a sinusoidal pattern as described by Equation (Equation 1), where $\psi_{l}$ represents the angular motion of the front servo, $\psi_{r}$ represents the angular motion of the rear servo, $\psi_{L0}-\psi_{R0}$ represents the phase difference between the front and rear servos, and $\theta_{L0}-\theta_{R0}$ represents the servo bias angle.
(1)$$\left\{\begin{array}{c} \psi_{l}=\psi_{L}\sin\left(2\pi ft+\psi_{L0}\right)+\theta_{L0}\\ \psi_{r}=\psi_{R}\sin\left(2\pi ft+\psi_{R0}\right)+\theta_{R0} \end{array}\right.$$

The undulation propulsion mode, depicted in Figure 5b, involves a 0.2 ms delay between the activation of the front and rear fin strips. The two fin strips have equal amplitudes and maintain a certain phase difference, resulting in periodic oscillations that drive the rubber membrane wing to create the undulating motion. The flapping propulsion mode, illustrated in Figure 5c, involves simultaneous activation of both fin strips. The front fin strip has a larger amplitude compared to the rear fin strip, resulting in a wave motion that gradually decreases from front to back, propelling the rubber membrane wing forward. As the main propulsion actuator in the MPF propulsion mode, the bio-inspired pectoral fin actuator generates stable and smooth thrust, providing the robotic manta ray with precise control forces for subtle adjustments during motion control. The designed motion of the bio-inspired robotic manta ray is achieved by four driving servos that actuate the two pairs of fin strips to perform cyclic oscillations. The rigid-flexible coupling design ensures the correct temporal sequence of pectoral fin motions while incorporating a certain level of passive flexibility to reduce resistance and increase radial force. The soft membrane wing undergoes passive deformation under the combined action of the active fin strips and water damping, generating a propulsion wave that propagates in the opposite direction, propelling the robotic manta ray forward. The front and rear pairs of fin strips enable precise control of the wave motion of the pectoral fin. Compared to other bio-inspired fish pectoral fins, the advantage of the proposed rigid-flexible coupling flapping structure lies in its ability to generate multiple motion modes, providing enhanced maneuverability. It also offers faster-flapping motion and greater flexibility in undulating movement, making it well-suited for a wide range of underwater tasks.

## 3. Recognition and Location Algorithms of Sea Cucumber Based on Improved YOLOv5s

### 3.1. YOLOv5s-ST Network

To improve the detection efficiency of sea cucumbers in practical applications and achieve real-time edge computing with a lightweight network, the YOLOv5s lightweight model is employed as the object detection model in this paper. The YOLOv5s model uses the CSPDarknet53 backbone, which, while stacking convolutional layers, widens the receptive field to capture local information and perform global information mapping based on the local information [40]. However, convolutional neural networks do not possess the same capability as transformers in extracting global feature information based on receptive fields and network depth [41].

To further extract global features from images and improve the accuracy of sea cucumber detection, this paper proposes a network based on YOLOv5s-ST (combining YOLOv5s with Swin Transformer). By introducing swin transformer blocks into the structure, it effectively considers shift invariance, scale invariance, and receptive field in convolutional neural networks, while also capturing global information and learning long-range dependencies. It ensures information propagation between windows through windows and shifted windows, effectively reducing the computational overhead in dense prediction tasks based on transformers. This achieves global modeling with good generalization capabilities [32]. The main improvement is the incorporation of swin transformer block modules into the two C3 modules of the CSPDarknet53 backbone, referred to as C3STR modules, to extract more advanced semantic features. This enhances the ability to extract globally correlated features from images. The YOLOv5s-ST algorithm framework is illustrated in Figure 6, and the structure of the swin transformer block is shown in Figure 7.

### 3.2. Sea Cucumber Positioning Method Based on Binocular Stereo Matching

This paper employs the HBV-1780-2S 2.0 model of a binocular camera, which captures left and right binocular images with a resolution of 640 × 480. The MATLAB Stereo Camera Calibrator toolbox is utilized for calibrating the parameters of the binocular camera. The stereo-matching process is implemented using the Semi-Global Block Matching (SGBM) algorithm.

The key steps for obtaining target information involve target recognition, with a focus on object keypoint detection and obtaining target position information. This paper combines the YOLOv5s-ST algorithm with binocular stereo-vision algorithms to achieve the localization and range of specific targets. The specific flowchart is illustrated in Figure 8. The target detection algorithm is capable of identifying the target’s category and center point coordinates in the image. Subsequently, the SGBM stereo-matching algorithm is employed to calculate the depth matrix of the target and obtain its three-dimensional coordinates. Finally, the distance to the target is computed, thereby achieving target localization and range.

## 4. Depth, Direction and Approach Control

Depth and direction control based on localization information is essential for the monitoring and operational tasks of the biomimetic robotic manta ray in aquaculture environments. Depth control ensures that the robotic manta ray maintains a specific depth in the water, allowing it to perform monitoring tasks within a designated depth range. This enables stable underwater footage, focuses on important scenes, and facilitates detailed inspection. direction control allows the robotic manta ray to move and monitor in specific directions, enabling comprehensive monitoring of aquaculture areas. Equipped with a variety of sensors, the biomimetic robotic manta ray can perceive its underwater state. By effectively integrating depth and direction control algorithms, the autonomy and flexibility of the robotic manta ray in water can be enhanced, enabling it to perform various tasks in complex underwater environments.

A fuzzy controller consists of four main components: fuzzification, fuzzy rule base, fuzzy inference, and defuzzification [42]. It is the core of a fuzzy control system. Its primary function is to map the input and output variables to membership functions and use a set of fuzzy rules based on empirical knowledge to determine the output. This improves the responsiveness and stability of the system [43].

In the depth control system, the depth sensor and the set depth value are the inputs to the controller. The error *e* and the rate of change in error *ec* are calculated based on these inputs. The error and error rate are used in the fuzzy PID controller to compute the modified values of the traditional PID parameters, namely $\Delta K_{p}$, $\Delta K_{i}$, $\Delta K_{d}$. Similarly, in directional control, the inputs to the fuzzy PID controller are the yaw angle error *e* and the rate of change in the yaw angle error *ec*. The fuzzy PID controller for the approach control consists of a depth controller and directional controller, which is illustrated in Figure 9. In the approach control system, the binocular camera sends the three-dimensional spatial coordinates of the sea cucumber to the lower-level controller. The lower-level controller utilizes the fuzzy PID controllers for depth and direction control to adjust the fin strip’s bias angle and amplitude for the next motion cycle.

In PID control, the initial values of three parameters, $K_{p}$, $K_{i}$, and $K_{d}$, need to be determined. These initial values can be determined using engineering measurement methods. Referring to Equation (Equation 2), the PID parameters are adjusted based on the correction information from the fuzzy PID controller.
(2)$$\left\{\begin{array}{c} K_{p}=K_{p0}+\Delta K_{p}\\ K_{i}=K_{i0}+\Delta K_{i}\\ K_{d}=K_{d0}+\Delta K_{d} \end{array}\right.,$$
$K_{p0}$, $K_{i0}$, and $K_{d0}$ represent the initial values of the PID controller, while $K_{p}$, $K_{i}$, and $K_{d}$ represent the adjusted output values.

The fuzzy inference designed in this study is based on the fuzzification of the error *e* and the error rate *ec*, as well as the fuzzy rule base, to derive the fuzzy subsets corresponding to $\Delta K_{p}$, $\Delta K_{i}$, and $\Delta K_{d}$. The three-dimensional surface plots of the fuzzy-inferred output variables are depicted in Figure 10. It can be observed that the output variables exhibit smooth changes as the input variables vary, which satisfies the basic requirements of fuzzy control rules. In the figure, different colors represent different values of $\Delta K_{p}$, $\Delta K_{i}$, and $\Delta K_{d}$, with yellow representing larger values and blue representing smaller values.

## 5. Experiments

### 5.1. Experiment and Analysis of Sea Cucumber Recognition and Location Algorithms Based on Improved YOLOv5s

The hardware environment used for model training in this study consisted of an Intel i7-1180H CPU and an Nvidia GeForce RTX3060 GPU. The software environment employed Python 3.6.5 and the PyTorch deep learning framework. The training process parameters are shown in Table 2.

Two sea cucumber datasets were used in this study to validate the model’s detection performance. The first dataset was from the China Underwater Robot Professional Contest (URPC2020) [44], which contains publicly available data. After data cleaning and partitioning, the sea cucumber dataset consisted of 3001 images with a total of 6808 ground truth bounding boxes. The training set contained 2370 images with 5413 ground truth bounding boxes, while the validation set consisted of 631 images with 1395 ground truth bounding boxes. The second sea cucumber dataset was collected from Mingbo Aquaculture Co., Ltd. (Yantai, China). The self-collected dataset was calibrated and divided, resulting in 630 sea cucumber images with a total of 1951 ground truth bounding boxes. The training set consisted of 490 images with 1510 ground truth bounding boxes, and the validation set comprised 140 images with 441 ground truth bounding boxes.

To better validate the effectiveness of the algorithm, this study conducted experiments on the URPC public dataset. In the same experimental environment, a comparative experiment and performance evaluation were performed using YOLOv5s-ST. Additionally, to address the challenges posed by harsh underwater environments, the relative global histogram stretching (RGHS) [45] image enhancement method was employed for preprocessing the images. The comparative results of model training on the public dataset are shown in Figure 11. From the figure, it could be observed that YOLOv5s-ST was capable of detecting smaller and more concealed sea cucumbers and successfully detecting a larger number of sea cucumbers. The comparison demonstrated that YOLOv5s-ST outperforms YOLOv5s in terms of detection performance.

As shown in Figure 12, the comparison of average precision (AP) before and after the improvement was depicted. The mAP@0.5 represents the AP for each category when the intersection over union (IoU) threshold is set to 0.5. It could be observed that the purple curve exhibited greater improvement compared to the yellow, green, and blue curves. The blue curve rises the slowest, indicating the slowest fitting speed during YOLOv4 training. Although the yellow and green curves have a faster-rising speed, with the increase in training rounds, the stable mAP@0.5 value is slightly lower than that of the purple and red curves, indicating that YOLOv5s and YOLOv7 have a faster fitting speed during training. However, the accuracy after full training is less than that of YOLOv5s-ST and YOLOv5s-ST-RGHS.

The specific training results are presented in Table 3. The calculations of Precision, Recall, and F1-Score are shown in Equations (Equation 3)–(Equation 5). In the formula, *TP* (True Positive) is the sample that is correctly predicted as sea cucumber; *FN* (False Negative) is the sample that is incorrectly predicted as the background; *TN* (True Negative) is the sample that is correctly predicted as the background; and *FP* (False Positive) is the sample that is incorrectly predicted as sea cucumber.
(3)Precision=TP/(TP+FP)
(4)Recall=TP/(TP+FN)
(5)$$F1\text{-}\mathrm{score}=\frac{2*Precision*Recall}{\left(Precision+Recall\right)}$$

According to the above table, it could be observed that the mAP@0.5 achieved 86.2% in the original YOLOv5s model. After improvement, the YOLOv5s-ST model achieved an average precision of 88.1%, showing an improvement of 1.9% compared to the original model. Finally, by applying RGHS for image enhancement on the dataset, the optimal experimental model of this study was obtained, with a mAP@0.5 of 88.4%. This represented a 2.2% improvement compared to the original model and demonstrated better performance compared to YOLOv4 and YOLOv7 training results, thus validating the effectiveness of the model improvement in this study.

For the self-collected dataset, comparative experiments were also conducted, and the detection result comparison is shown in Figure 13. The mAP@0.5 was 93.5% for the YOLOv5s. In this study, the model trained on the URPC dataset was used as the pre-trained model for the self-collected dataset. By training with YOLOv5s-ST, the mAP@0.5 improved to 94.3%, showing a relative improvement of 0.8% compared to the original model. Finally, by applying RGHS for image enhancement on the dataset, the optimal experimental model of this study was obtained, with an average precision mAP@0.5 of 94.5%, representing a 1.0% improvement compared to the original model. These results met the experimental requirements and demonstrated better performance compared to the training results of YOLOv4 and YOLOv7. The training process and specific experimental results are shown in Figure 14 and Table 4. It can be seen from Figure 14 that YOLOv4 has a slow convergence rate during training. Although the mAP@0.5 value of YOLOv5s and YOLOv7 is nearly the same as that of YOLOv5s-ST and YOLOv5s-ST-RGHS, their curves are not as stable as those of YOLOv5s-ST-RGHS. The results show that YOLOv5s-ST-RGHS has better model stability and detection accuracy.

### 5.2. Experiment and Analysis of Binocular Positioning

To test the localization accuracy of the binocular positioning method, distance measurements were taken every 40 cm. Manual methods were used to measure the positioning of the binocular camera relative to the sea cucumber. This experiment was conducted in a recirculating water tank in the fish farming facility. Since the sea cucumbers adhered to the bottom of the tank, the binocular camera was moved during the experiment. The movement may have caused slight variations in the horizontal and vertical directions. Figure 15 illustrates the target recognition and 3D positioning of a single sea cucumber using the binocular camera. By performing 3D positioning and distance measurements on individual targets, the localization accuracy of the aforementioned method was verified. The measurement results are shown in Table 5.

Based on the underwater experiments, the computer vision-based sea cucumber recognition and localization algorithm studied in this paper could accurately locate sea cucumbers in underwater environments. Based on the calculated data from the experiments, the average relative error in sea cucumber location was 1.97%, with a maximum relative error of 4.21%. This indicated that the proposed method can effectively and accurately localize sea cucumbers, providing reliable target position information for underwater robots.

### 5.3. Experiment and Analysis of Depth, Direction, and Approach Control

To validate the performance of the depth control, direction control, and approach control of the biomimetic robotic manta ray, a variety of experiments were conducted in an open water environment using the established motion control system. The objective of these experiments was to demonstrate the effectiveness of the depth, direction, and approach control of the system.

#### 5.3.1. Experimental Scheme

The experimental setup for the motion control of the biomimetic robotic manta ray in this study primarily consisted of the biomimetic robotic manta ray and a remote control terminal. The two components communicated and transmitted data through a wireless RF module. Multiple motion control experiments were conducted at the sedimentation pool of Mingbo Aquaculture Company in Laizhou, China. The experimental site is depicted in Figure 16.

In this paper, the Jetson Xavier NX edge computing box was used to realize target detection and binocular stereo matching, and the three-dimensional coordinate information of the sea cucumber target was calculated. The binocular camera transmitted the image data to Jetson Xavier NX for processing, calculated the three-dimensional coordinate information of the target, and transmitted it to the STM32F407 master controller through serial communication. The STM32F407 controlled the bionic manta ray’s movement based on the real-time transmission of target coordinate information.

#### 5.3.2. Experimental Result

In the depth control experiment, the biomimetic robotic manta ray had an initial depth of 0.2 m, and the target depth was 1 m. Figure 17 illustrates the depth variation curve, where the brown dashed line represents the desired depth and the blue solid line represents the actual depth measured by the sensor. The biomimetic robotic manta ray reached the target depth within 7 s and maintained it in the vicinity of the target depth. Several snapshots of the closed-loop depth control experiment are shown in Figure 18. The biomimetic robotic manta ray rapidly reached the desired depth with no significant overshoot and remained stable near the desired depth. The steady-state error was within the range of (−30 mm, 30 mm). This indicated that the depth controller was capable of achieving precise depth control for the biomimetic robotic manta ray.

During the motion of the biomimetic robotic manta ray, there was a risk of collision with obstacles due to the absence of active obstacle avoidance capability. This can cause loosening or deformation of the pectoral fin linkage mechanism, resulting in inconsistent thrust generated by the left and right pectoral fins, thereby affecting the motion posture of the biomimetic robotic manta ray. Therefore, a direction control system was designed to adjust the motion posture of the biomimetic robotic manta ray by altering the flapping amplitude of the left and right pectoral fins. Figure 19 illustrates the yaw angle variation curve (blue solid line) under the influence of the direction control system during straight swimming, with the target yaw angle indicated by the orange dashed line. The biomimetic robotic manta ray reached the target angle within 2 s and maintained it in the vicinity of the target angle. The experimental results demonstrated that, under the effect of the direction control system, the biomimetic robotic manta ray could maintain the desired yaw angle with a steady-state error within the range of (−1°, 1°).

The rapid descent process of the biomimetic robotic manta ray resulted in large changes in the viewing angle, leading to target loss. Since the prototype lacks a gimbal system, it only performed two-dimensional approach motions. The screenshots of the approach motion and underwater sea cucumber localization video sequences are shown in Figure 20. From 0 to 6 s, the manta ray performed the approach motion and then started to move away after 6 s. Once the biomimetic robotic manta ray detected the three-dimensional spatial coordinates of the sea cucumber, it fine-tuned its motion direction by adjusting the deflection angles of the left and right pectoral fins. In the underwater images, the midpoint position of the left boundary was taken as the origin (0, 0) for the coordinate system. The x-values to the right of the origin were all greater than 0, while the y-values above the origin were negative, and below the origin were positive. The two-dimensional coordinate changes of the sea cucumber are illustrated in Figure 21. In the y-direction, the distance between the biomimetic robotic manta ray and the sea cucumber initially decreased and then increased. In the x-direction, at 2nd, the biomimetic robotic manta ray adjusted its motion direction toward the sea cucumber target, and at 5 seconds, it further adjusted its motion direction. At 6 seconds, it crossed over the sea cucumber from above and gradually moved away. The distance variation curve between the biomimetic robotic manta ray and the sea cucumber is depicted in Figure 22, showing that the distance initially decreased and then increased, indicating that the biomimetic robotic manta ray first approached and then moved away from the target under the effect of inertia. This confirmed that the biomimetic robotic manta ray could perform approach motions toward the sea cucumber target under the influence of the approach control system.

## 6. Discussion

The proposed YOLOv5-ST object detection algorithm in this paper enhances the model’s global feature extraction capability by introducing swin transformer blocks. The model comparison experiments in Table 3 and Table 4 also demonstrate the effectiveness of the improved model, achieving high detection accuracy. Although the model introduces a small number of transformer blocks, it slightly increases the computational cost of the model. However, this increase has minimal impact on the overall computational cost of the model. Since object localization requires the integration of binocular stereo-matching algorithms, it occupies significant computational resources, affecting the real-time performance of the overall detection and localization algorithm. To improve the real-time capability of localization, it is necessary to reduce the computational cost of stereo matching by improving binocular stereo-matching algorithms or using local image stereo-matching techniques [20] and other methods. Compared with the YOLOv4 algorithm, the YOLOv5 algorithm itself has higher algorithm accuracy and model performance. Although the accuracy of the YOLOv7 algorithm has decreased compared with YOLOV5-ST, considering the superiority of its algorithm itself, there is still room for improvement to achieve higher detection accuracy.

The biomimetic manta ray robot designed in this paper generates thrust by periodically oscillating two pairs of fins with specific amplitudes and frequencies. By changing the rotation angles of the servos and the phase difference between the two fins, the robot can perform various modes of motion. Its rigid-flexible coupling design has advantages [13,46], where passive deformation of the soft wing generates effective propulsion, allowing for the replication of the complex flexible deformation of real batoids while ensuring the lightweight and flexibility of the overall mechanism. Although the swimming of the biomimetic manta ray is relatively stable, it still affects the image quality and the success rate of binocular stereo matching in underwater image acquisition. Therefore, optimizing the control algorithm is needed to achieve smoother motion for the manta ray. The depth, direction, and approach control experiments of the manta ray demonstrated effective detection and localization of sea cucumbers in aquaculture ponds. However, the patrol path of the manta ray exhibits randomness, representing an initial exploration of applying underwater biomimetic robots to the aquaculture industry. It is necessary to implement global path-tracking control for the manta ray to improve its ability to traverse and monitor, enabling efficient applications in underwater biological monitoring and empowering aquaculture.

## 7. Conclusions

In this paper, we have designed and implemented a small biomimetic manta ray robot with visual perception capabilities and a rigid-flexible coupled pectoral fin for sea cucumber recognition, localization, and approach. First, the mechanical structure of the manta ray robot was designed as a platform for subsequent underwater monitoring. Second, by improving the YOLOv5 object detection algorithm and integrating it with binocular stereo matching, precise sea cucumber identification, and localization were achieved. Finally, a fuzzy PID controller was designed to realize depth control, direction control, and target approach motion control for the manta ray robot. Experimental results demonstrate that the improved YOLOv5 object detection algorithm achieves detection accuracies (mAP@0.5) of 88.4% and 94.5% on the URPC public dataset and self-collected dataset, respectively, effectively recognizing and localizing sea cucumbers. Control experiments were conducted, validating the effectiveness of the robotic manta ray’s motion toward sea cucumbers. Experimental results confirmed the usability of the manta ray platform, the accuracy of the improved algorithms, and the effectiveness of the approach motion. This work provides valuable insights for the future development of more intelligent and efficient underwater biomimetic detection platforms, offering a novel solution for intelligent monitoring in aquaculture.

## Figures and Tables

**Figure 1 biomimetics-08-00345-f001:**
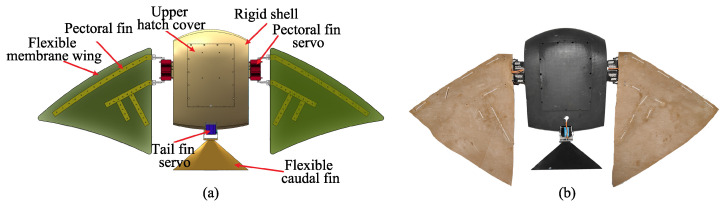
The design drawings and prototype of the robotic manta ray. (**a**) the overall rendering of the robotic manta ray. (**b**) the prototype of the robotic manta ray.

**Figure 2 biomimetics-08-00345-f002:**
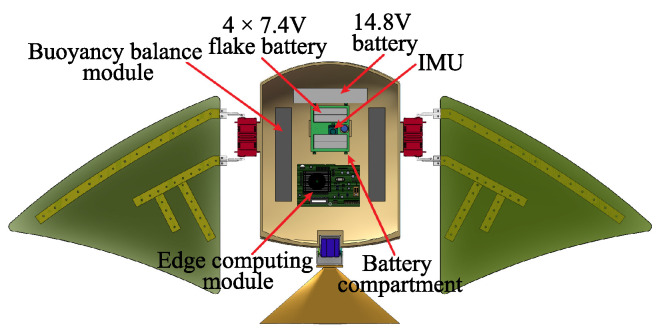
The interior layout of robotic manta ray.

**Figure 3 biomimetics-08-00345-f003:**
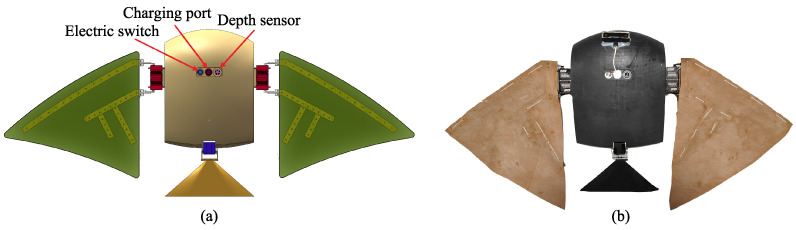
The bottom layout of the robotic manta ray. (**a**) The overall rendering of the robotic manta ray. (**b**) The prototype of the robotic manta ray.

**Figure 4 biomimetics-08-00345-f004:**
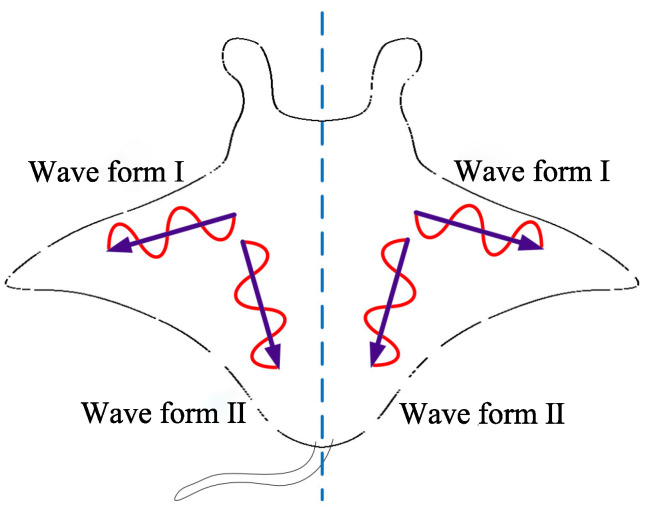
Undulation analysis of manta ray.

**Figure 5 biomimetics-08-00345-f005:**
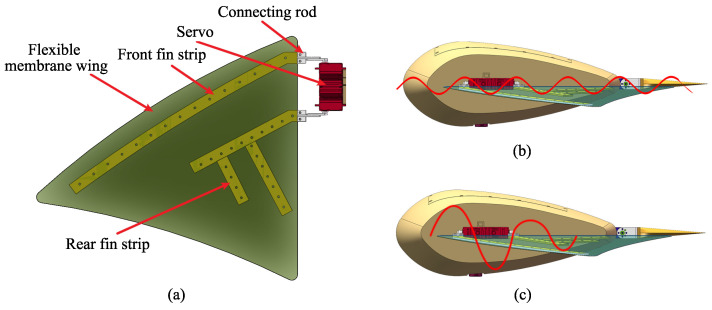
The construction of the pectoral fin. (**a**) Overall structure of the pectoral fin. (**b**) Schematic diagram of undulation propulsion mode. (**c**) Schematic diagram of flapping propulsion mode.

**Figure 6 biomimetics-08-00345-f006:**
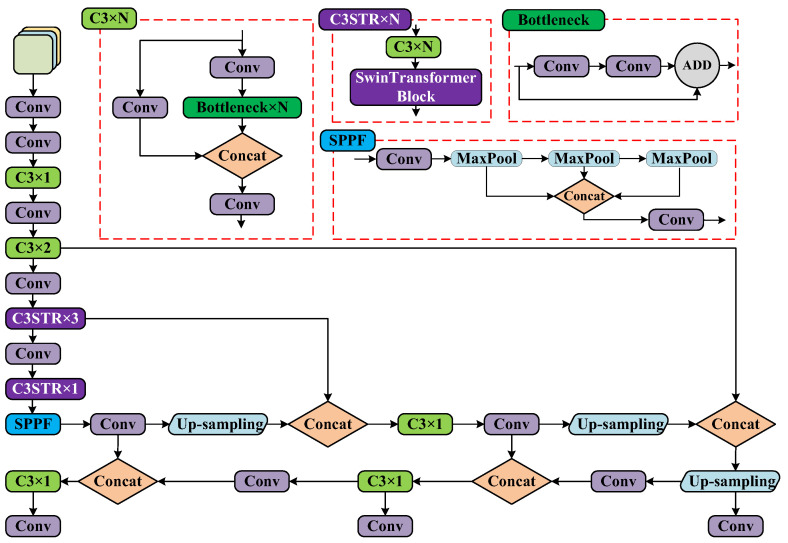
The diagram of the YOLOv5s-ST network structure.

**Figure 7 biomimetics-08-00345-f007:**
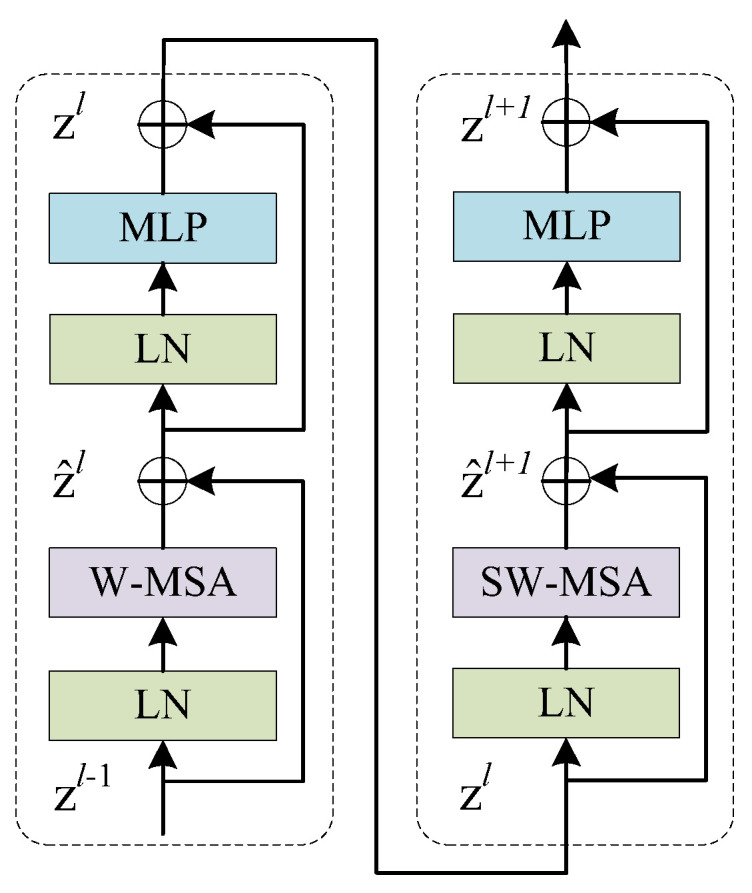
Swin Transformer Block.

**Figure 8 biomimetics-08-00345-f008:**
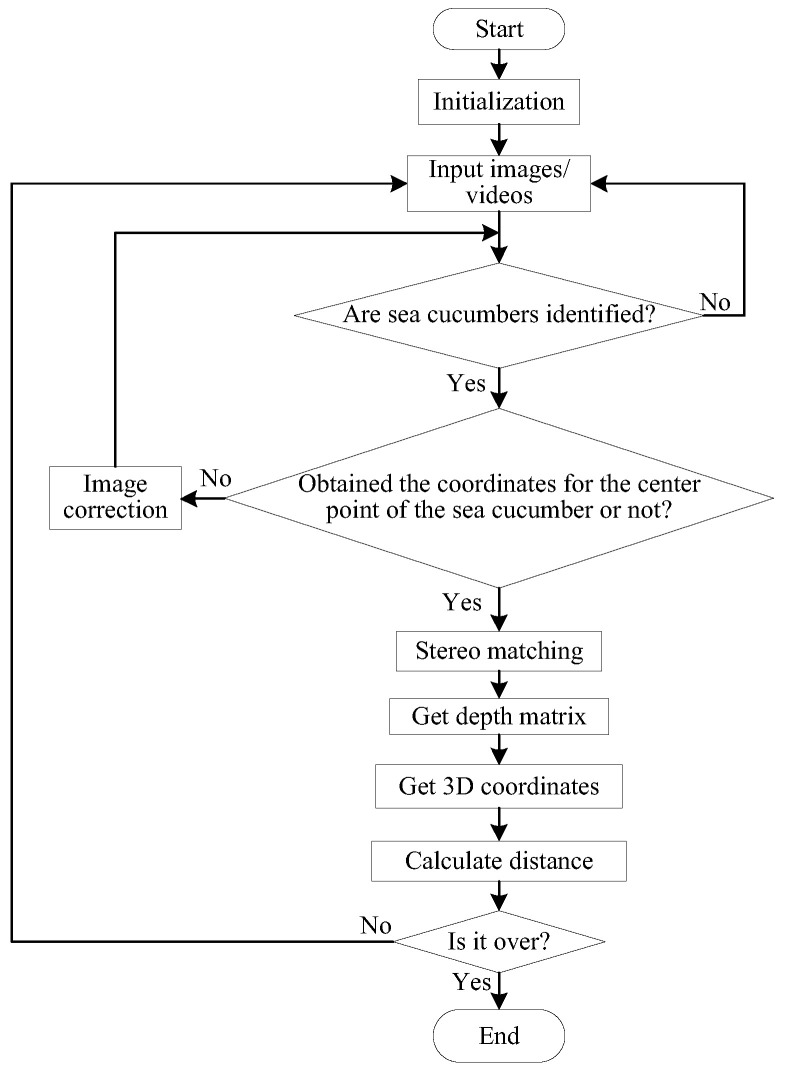
The flowchart of binocular positioning.

**Figure 9 biomimetics-08-00345-f009:**
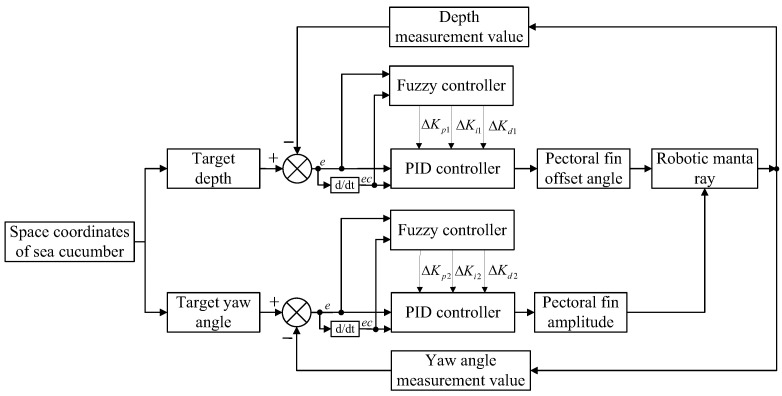
The structure of the approach controller.

**Figure 10 biomimetics-08-00345-f010:**
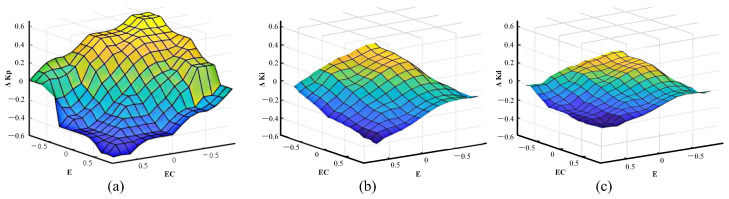
Fuzzy output. (**a**) The three-dimensional surface plots of $\Delta K_{p}$. (**b**) The three-dimensional surface plots of $\Delta K_{i}$. (**c**) The three-dimensional surface plots of $\Delta K_{d}$.

**Figure 11 biomimetics-08-00345-f011:**
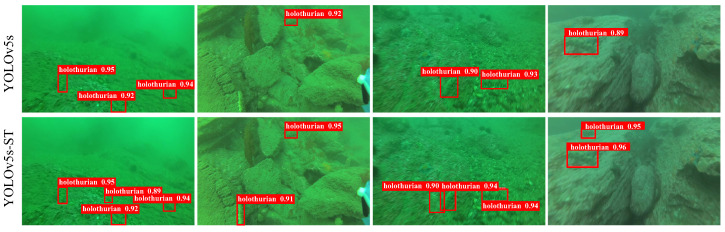
Comparison of detection results of URPC datasets.

**Figure 12 biomimetics-08-00345-f012:**
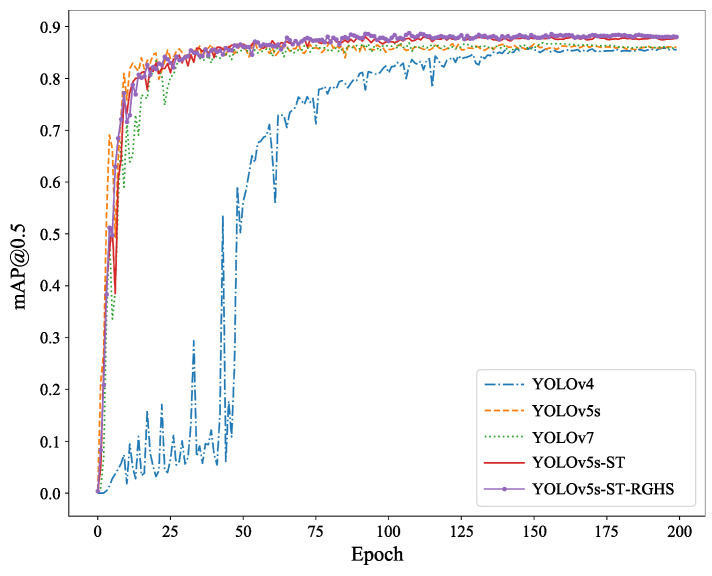
Comparison chart of training results of URPC dataset.

**Figure 13 biomimetics-08-00345-f013:**
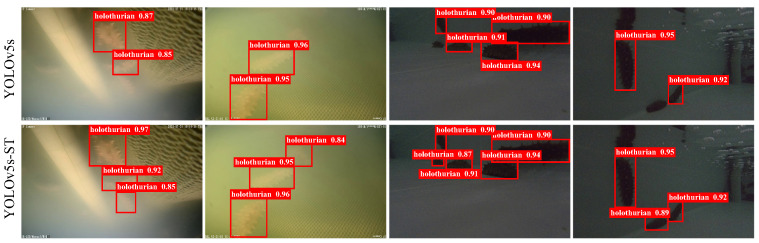
Comparison of detection results of self-collected datasets.

**Figure 14 biomimetics-08-00345-f014:**
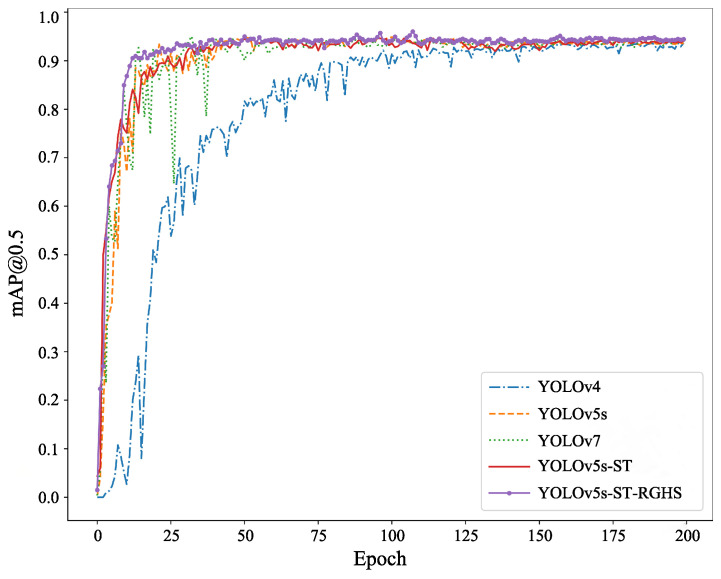
Comparison chart of training results of self-collected dataset.

**Figure 15 biomimetics-08-00345-f015:**
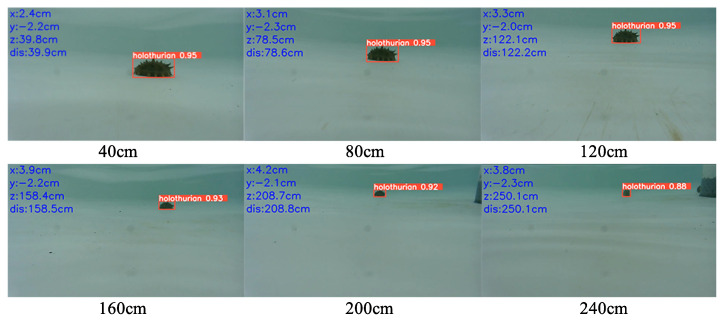
The experiment of binocular positioning.

**Figure 16 biomimetics-08-00345-f016:**
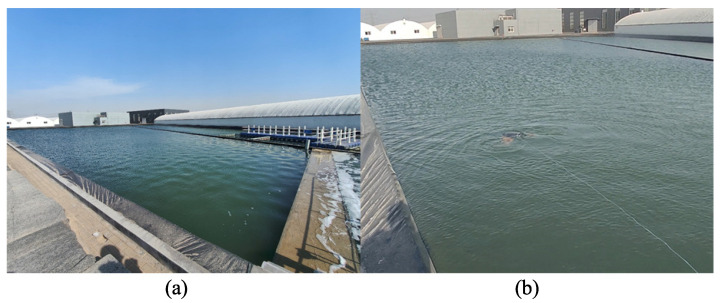
The experiment environment of motion control. (**a**) Experimental environment. (**b**) The scenario where the robot is swimming in the open pool.

**Figure 17 biomimetics-08-00345-f017:**
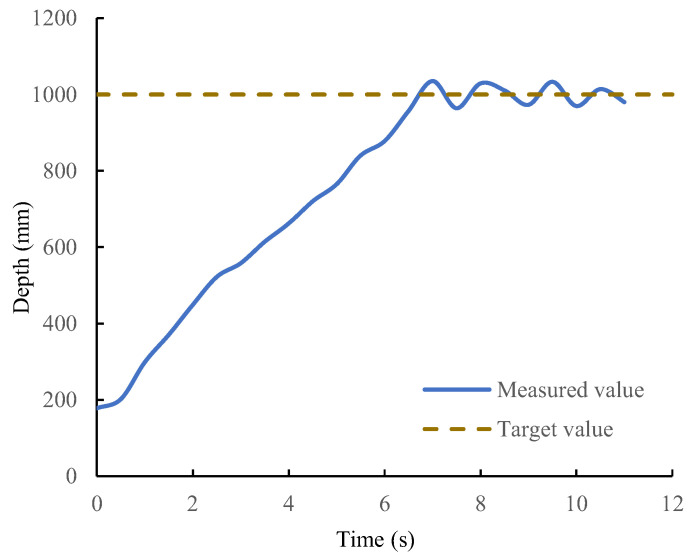
Depth change curve of the closed-loop depth control.

**Figure 18 biomimetics-08-00345-f018:**
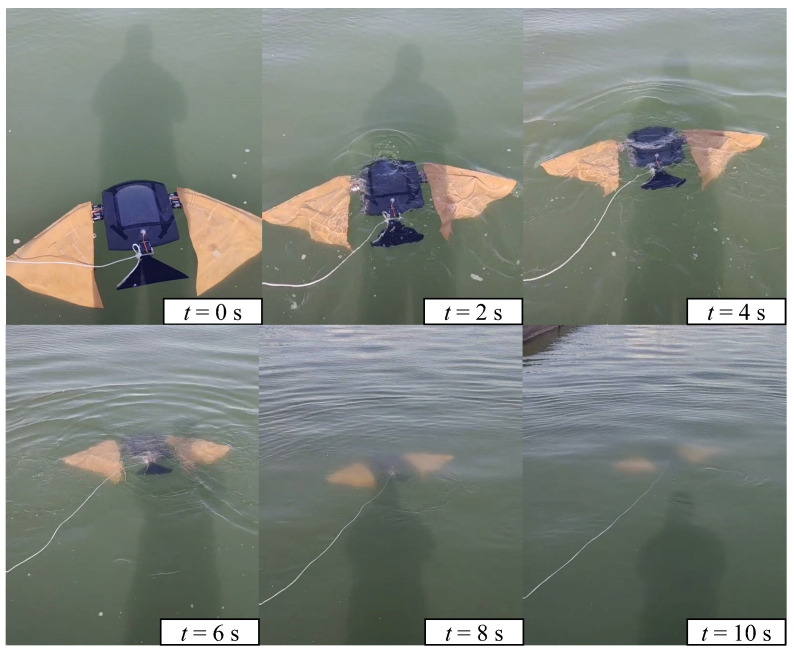
Snapshots of the depth control for the robotic manta ray.

**Figure 19 biomimetics-08-00345-f019:**
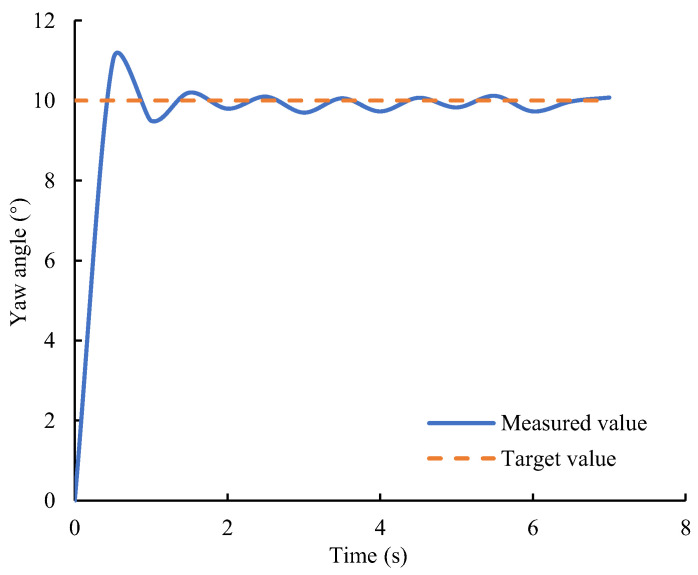
Yaw angle change curve of direction control.

**Figure 20 biomimetics-08-00345-f020:**
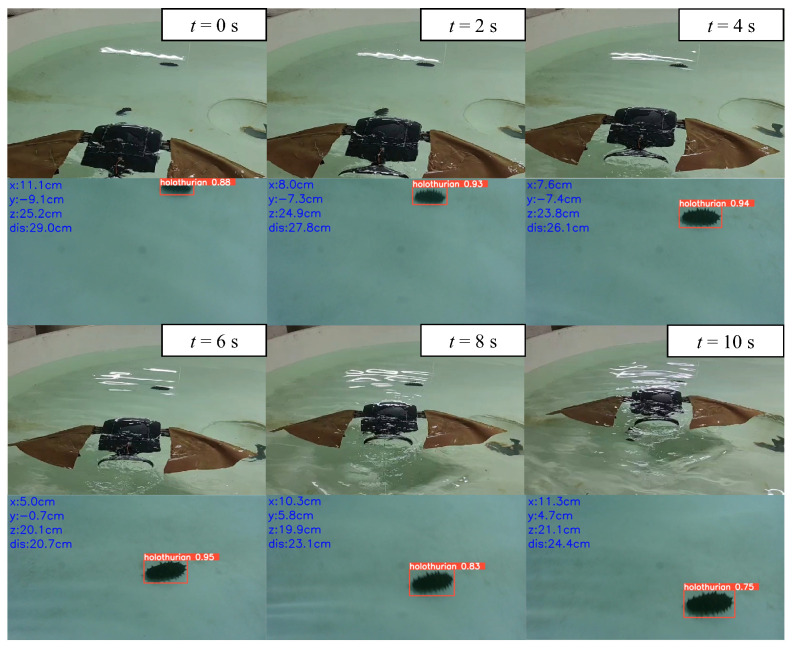
The screenshots of the approach motion and underwater sea cucumber localization video sequences.

**Figure 21 biomimetics-08-00345-f021:**
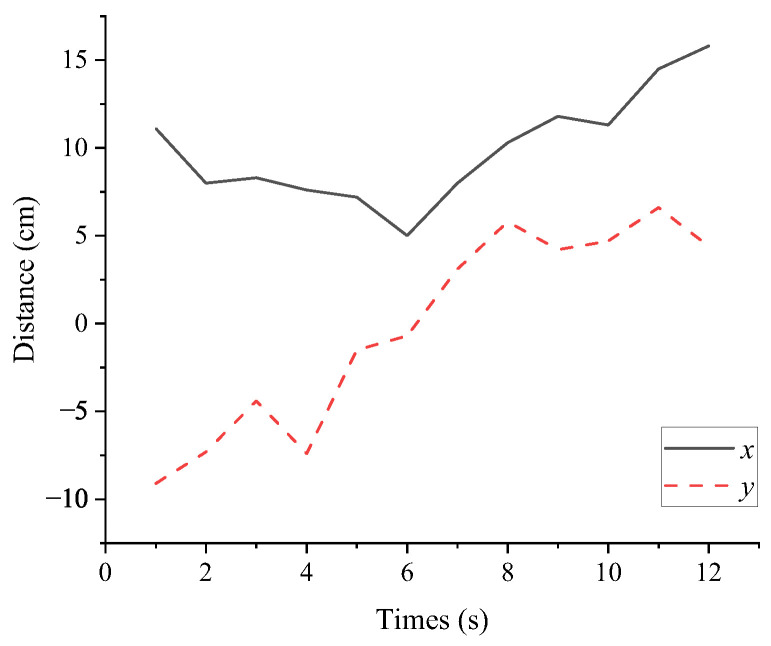
The two-dimensional coordinate changes of the sea cucumber.

**Figure 22 biomimetics-08-00345-f022:**
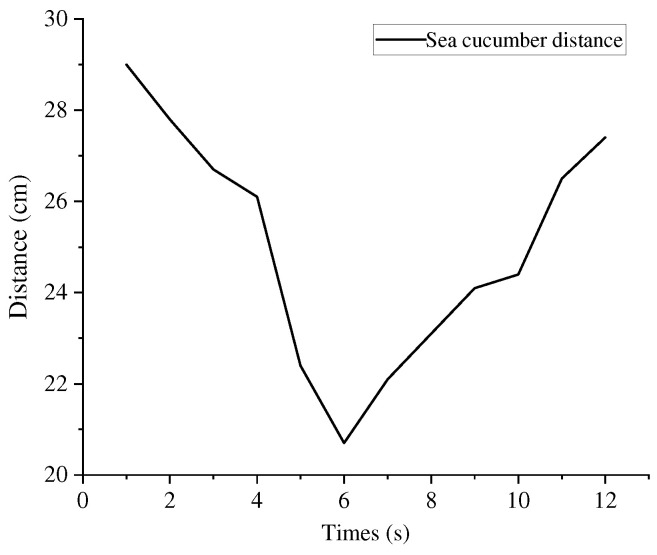
The distance variation curve between robotic manta ray and sea cucumber.

**Table 1 biomimetics-08-00345-t001:** The technical parameters of robotic manta ray.

Technical Parameters	Values
Size	560 mm × 1300 mm × 120 mm
Weight	12.25 kg
Body shell	Black resin
Pectoral fin	Spring steel, rubber mold (1 mm)
Caudal fin	Rubber (Shore hardness 40)
Main controller	STM32F407
Power supply	DC 7.4 V/14.8 V
Working hours	3 h
Sensor	Depth sensor, IMU, power metering sensor
Inching switch	YJ-GQ22AF
Charging port	M12

**Table 2 biomimetics-08-00345-t002:** The parameters of the training process.

Parameters	Values
Input image	640 × 640
Class number	1
Batch size	16
Learning rate	0.001
Momentum factor	0.95
Weight decay coefficient	0.001
Iterations	200

**Table 3 biomimetics-08-00345-t003:** Comparison table of training results of URPC dataset.

	Precision	Recall	F1-Score	mAP@0.5
YOLOv4	0.576	0.867	0.692	0.855
YOLOv5s	0.881	0.778	0.826	0.862
YOLOv7	0.877	0.809	0.842	0.858
YOLOv5s-ST	0.879	0.798	0.837	0.881
YOLOv5s-ST-RGHS	0.882	0.801	0.840	0.884

**Table 4 biomimetics-08-00345-t004:** Comparison table of training results of self-collected dataset.

	Precision	Recall	F1-Score	mAP@0.5
YOLOv4	0.665	0.950	0.782	0.937
YOLOv5s	0.912	0.877	0.894	0.935
YOLOv7	0.868	0.892	0.880	0.939
YOLOv5s-ST	0.925	0.871	0.897	0.943
YOLOv5s-ST-RGHS	0.907	0.891	0.899	0.945

**Table 5 biomimetics-08-00345-t005:** Comparison of experimental results of binocular positioning.

Experimental Distance (cm)	3D Coordinates (cm)	Measured Distance (cm)	Error (%)
40	(2.4, −2.2, 39.8)	39.9	0.25
80	(3.1, −2.3, 78.5)	78.6	1.75
120	(3.3, −2.0, 122.1)	122.2	1.83
160	(3.9, −2.2, 158.4)	158.5	0.94
200	(4.2, −2.1, 208.7)	208.8	2.85
240	(3.8, −2.3, 250.1)	250.1	4.21

## Data Availability

The data presented in this study are available on request from the corresponding author.

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
