# Peer review of "Design and Realization of a Novel Robotic Manta Ray for Sea Cucumber Recognition, Location, and Approach"

_biomimetics, 2023, doi:10.3390/biomimetics8040345_

Round 1
Reviewer 1 Report
The article focuses on the design of a robotic manta ray for sea cucumber recognition, localization, and approach. Initially, a prototype of the robotic manta is developed. Then, object detection is performed using the YoloV5 algorithm. A motion controller is proposed for autonomous 3D monitoring tasks. Finally, the capabilities of the robot are validated through various measurements. The article is well-written; however, the following suggestions may improve the quality of the publication:
The abstract should briefly highlight the research problem, existing solutions, and research gap.
The choice of keywords should be more relevant to the research.
In Sections 2, 3, and 4, fewer references are used. It is suggested to justify statements with suitable references.
In line 243, provide the full form of the abbreviation followed by the abbreviation inside the brackets.
In Figure 9, the output of the "Robotic manta ray" is going outside.
In Figure 10, different scales are used in one dimension, making it difficult to compare.
Different flavors of Yolo algorithm are used in the research, while Yolo v5 is only under discussion, please clarify.
The discussion section only targets the Yolo v5 algorithm, ignoring other techniques. It is suggested to provide an overview of the techniques used in the research with their pros and cons.
The conclusion should focus more on the results and their significance.
Minor editing of English language required
Reviewer 2 Report
The sea agriculture is extremely developing field today. The main negative factor that influence on its development is difficulties related with control and harvesting of products. In presented paper authors demonstrates very interesting approach based on the application of bioinspired robotic manta ray as harvester for sea cucumber. Presented robot made from available components and have significant practical perspectives. Recognition system also demonstrates good results in the searching of sea cucumber. Manuscript is well written, research design is solid only minor revision is needed. Main comments are:
1. Abstract should contain information concerning results of application presented in the paper robotic manta ray;
2. Figure 2 and 3, please make all letters a little bit bigger, in current state it is difficult to read;
3. Line 302, reference to the publicly available data from URPC2020 is needed;
Moderate check of English langauage is necessary;
Reviewer 3 Report
The paper has been well-written. Some comments and suggestions are recommended. Please see the attached file.

The quality of English is good. Please check typing and grammar errors carefully.
